# Blockchain for Governments: The Case of the Dubai Government

Shafaq Khan [1], Mohammed Shael [2], Munir Majdalawieh [3,*], Nishara Nizamuddin [3] and Mathew Nicho [3]

[1] School of Computer Science, University of Windsor, Windsor, ON N9B 3P4, Canada; shafaq.khan@uwindsor.ca

[2] Department of Economic Development, Dubai P.O. Box 13223, United Arab Emirates; mohammed.shael@dubaided.gov.ae

[3] College of Technological Innovation, Zayed University, Dubai P.O. Box 19282, United Arab Emirates; nishara.nizamuddin@zu.ac.ae (N.N.); mathew.nicho@zu.ac.ae (M.N.)

[*] Correspondence: munir.majdalawieh@zu.ac.ae; Tel.: +971-4402-1393

**Abstract:** Blockchain technology is an innovative technology with the potential of transforming cities by augmenting the building of resilient societies and enabling the emergence of more transparent and accountable governments. To understand the capabilities of blockchain, as well as its impact on the public sector, this study conducted a review of blockchain technology and its implementations by various governments around the globe. E-government evolution is analyzed, based on empirical evidence from a Dubai government entity in the United Arab Emirates (UAE), which has utilized blockchain technology for developing end-user services, relevant to the public sector. Benefits achieved and challenges to overcome in such blockchain-based pilot deployments are discussed. The findings of this study offer new insights for practitioners involved in bringing in innovations for the benefit of society, using blockchain technology. Furthermore, it provides insights into policy actions to be developed to address the future challenges and to improve already existing e-government policies. The results of this research will benefit all blockchain-based pilot deployments by providing guidance and knowledge on this immature yet developing technology.

**Keywords:** blockchain; e-government services; department of economic development; e-government policies; end-user services

## 1. Introduction

Electronic government (e-government) defines the existing and future interactions between the government and its citizens [1–3]. Good quality e-services provided by private sectors are making people less tolerant of poor public sector services. Consequently, the concept of e-government to enhance quality of life is gaining increasing importance, and public e-services have become an increasingly important topic on the public agenda [4]. The outbreak of the COVID-19 pandemic has even catalyzed e-government processes [5,6], turning them into a necessity, rather than an ambition to improve the quality of life [7]. Consequently, it is in every government's interest to deliver more transparent, efficient, and effective public services to citizens. By doing so, governments can improve their performance, create new public value for citizens and businesses, and bring in better quality, efficiency, effectiveness, and transparency in the administration of the public state institutions [2]. Such evolvement culminates in better collaboration with partners of all types, be it other government or private organizations. Besides offering the improvement in processes at reduced costs [8], it also reduces the administrative workloads for government employees [9]. Blockchain implementation has emerged as a priority for several governments to innovate their technology and digital service landscape. Consequently, several governments, such as the Netherlands, Denmark [10], Saudi Arabia [11], and the United Arab

Emirates (UAE), have already made e-government services mandatory [12], irrespective of the (un) skills of the citizens or irrespective of the complexity of the services [1,10].

By integrating blockchain, artificial intelligence, and big data technologies in their processes, governments are moving towards data-driven and evidence-based decisions and policymaking. This helps governments in promoting sustainable development to support resilient societies [13,14]. While the world is witnessing rapid adoption of blockchain technology, with an expected market value of more than USD 176 billion by 2025 [15], its application in e-government is also gaining ground [16]. Blockchain-enabled governments can transform government processes, achieve higher levels of efficiency, and improve the management of the public benefits [17,18]. Blockchain technology can be used as a platform to transform the e-business operating models to offer fully integrated services and enforce common business rules [13]. This technology can benefit the government and the public sectors in certifying identities, establishing trust, exchanging assets between parties across borders, and sealing digital contracts [19]. Blockchains are secure and "an open, distributed ledger that can record transactions between two parties efficiently and in a verifiable and permanent way" [20]. These benefits can augment the building of resilient societies, by keeping track of data across various activities and actors, authenticating and guaranteeing the execution of tasks, and enabling the emergence of more transparent and accountable governments [14].

The blockchain supported by governments in emerging markets has the potential to provide security and privacy of data with a high degree of sensitivity in addition to solving problems related to control over information and access [21,22].

There are many examples of successful e-government implementations using blockchain technology. For the Republic of Moldova, it has contributed to an increased inward, capital investment flows, and reduction of corruption practices [23] and Estonia is one of the world's leading governments in securing public health records with a form of Blockchain technology [24]. While governments in countries, such as the USA, Sweden, Singapore, and the UK, are leading in blockchain adoption for e-government services, every country needs to be concerned about it [25] and needs to investigate the potential use of blockchain technology as a platform for various applications in e-government [26]. The cryptocurrencies of blockchain technology have been in focus and adopted in many applications; however, attention to the underlying blockchain technology has been ignored. The blockchain technology that is disrupting private, as well as government sectors, is explored by a growing number of researchers worldwide. However, there are gaps found in blockchain adoption in the context of e-government that are unexplored in academic literature [27]. More specifically, there is a lack of research that is essentially focused on understanding how government services and operations can be improved through the use of blockchain technology, through the prism of specific case studies [28]. Researchers have also identified the lack of policy frameworks as one of the obstacles to implementing this technology across the globe by governments [29]. In addition, a large gap exists between the cultural and social characteristics of developing countries, including the Arab nations, from those of the Western nations [1]. Prior research on e-government adoption has largely focused on the developed countries, with only a few studies on the Arab region specifically [30,31]. The need for awareness and an increased adoption of this technology in both developed and developing countries is of paramount importance. Consequently, this study focuses on addressing the following three questions:

Q1: What are governments currently doing with blockchain technology?

Q2: What benefits and challenges does blockchain technology bring to digital governments?

Q3: What policy actions are needed to fully utilize these technologies for the benefit of society?

Section 2 begins by providing motivations for organizations to adopt blockchain technology. Further, it examines and throws light on some of the active blockchain-based national and local level government projects. Section 3 provides contextualization of blockchain technology by discussing potential benefits and technological hurdles related to its implementation. The methodology is discussed in Section 4. Section 5 discusses the Case Study Assessment Framework in terms of the pilot deployment at the Dubai Economic

Department (DED). While Section 6 lists the insights from our empirical study, Section 7 concludes by highlighting the policy actions that are required to support the full utilization of this technology for the benefit of society. Future work of this study is also highlighted in this section.

## 2. Blockchain and Digital Governments

There is a multitude of motivations for organizations to adopt blockchain technology in comparison to the traditional databases. The contents of centralized databases are stored in the centralized servers or databases in an organization to which any malicious user will be able to gain access to the system and may corrupt or destroy the information. From the moment data or information is stored on a centralized database, a dependency on a central database and administrator is created which leads to security breaches. For instance, the National Health Service (NHS) suffered coding errors for 150,000 patients in England when it was involved in a data breach, exposing sensitive medical data without patients' permission or knowledge [32]. A problem with the software used by physicians to record medical treatment details was identified. Patients had only agreed for the data to be used for individual care; however, the data were exposed to external auditors and for research [32]. Several other targeted attacks, such as ransomware, botnets, and Distributed Denial of Service (DDoS) have become rampant, causing great damage to the e-services [33]. With blockchain, the scenario is different, as it enables us to keep information that is not just relevant at a given time but also stores details of all the transactions and information that has taken place in history [34]. When hospitals and banks which deal with sensitive information store their data on a central database, in case of a data breach, the data become corrupted or locked, which leads to detrimental effects. With blockchain, all the nodes within the network are notified about the activity in the network, ensuring the integrity of the data.

Over the last two decades, there has been a persistent positive trend towards higher levels of e-government developments globally. Governments around the world have been proactively initiating widespread innovation and digital transformations across multiple levels and platforms. Governments have realized its far-reaching potential, not just for greater efficacy and effectiveness of the public service delivery, but also for ensuring inclusion, participation, and accountability. Figure 1 shows the utilization of blockchain for government sector services around the globe. The government of India is aiming to launch a smart warfare solution by adopting the Distributed Ledger Technology (DLT) to safeguard the safety and security of the critical infrastructure. The central bank of India has shown immense interest in developing digital rupees and implementing blockchain technology for land registry systems [35–37]. The Government of Canada (GC) has been experimenting with utilizing blockchain innovation to give venture-based representatives a sort of advanced CV, giving "a changeless, self-possessed and secure record of their abilities and experiences" [38].

Sweden's government's land-ownership authority, "the Lantmäteriet", is aiming to launch its first blockchain technology property transaction for land registrations. The case study submitted by McMurren et al. [39] for addressing transaction costs through blockchain and identity in Swedish land transfers states that processing a land deal right from signing a contract until the completion of land transfer approximately takes four months. Signing a contract might take 2 h as verifying documents and identity management is performed manually. Moreover, this land transfer process not being completely digital leads to missing, incomplete, or wrong entries in the land registry [39]. Furthermore, the security structure of legacy systems prevents digitization because of the firewalls or limitations on network connections where organizations, such as government regulatory agencies, banks, and realtors, could gain access to the system, but not the sellers, and consumers.

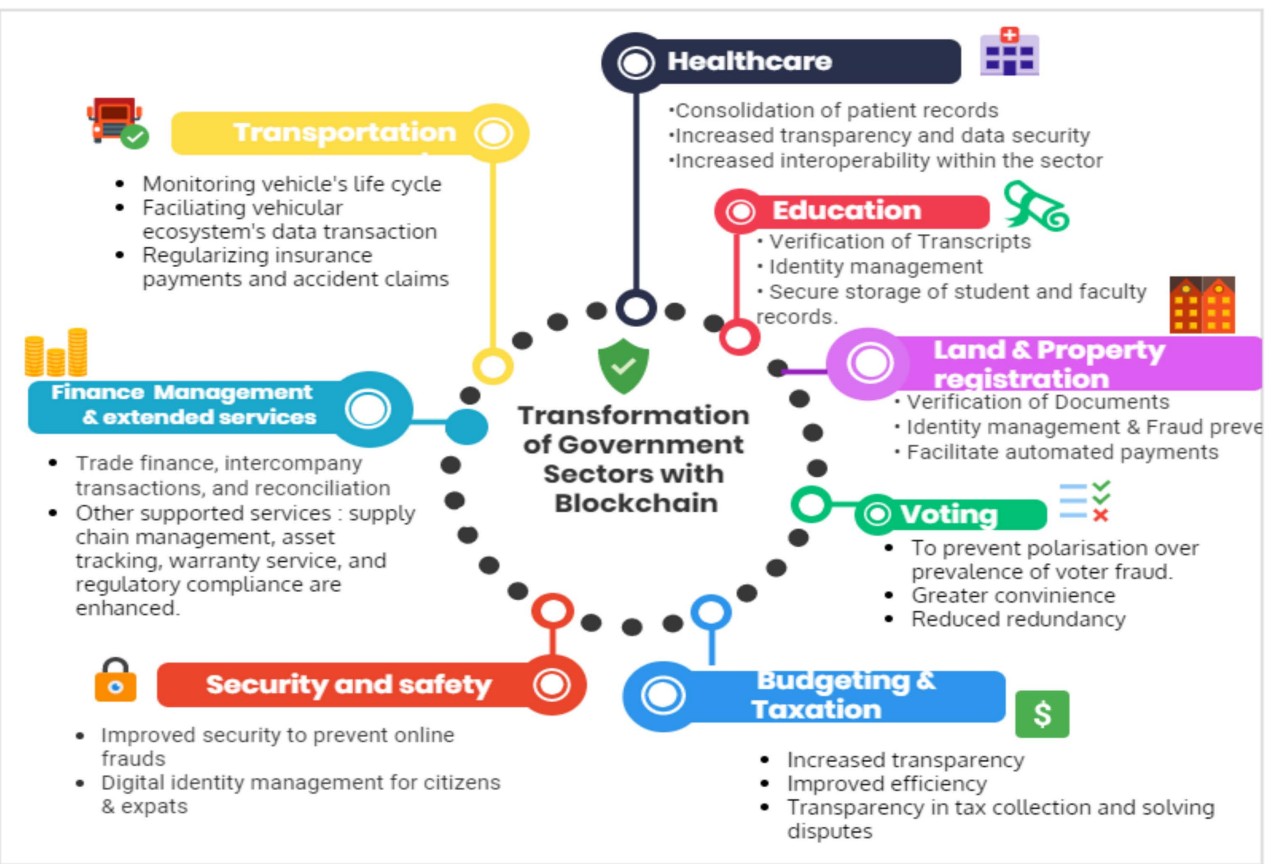

**Figure 1.** Utilizing Blockchain for Government sector services.

Malta has established three cryptocurrency and blockchain related laws, Liechtenstein has proposed a blockchain act enabling a "token economy". The UK is automating the process of regulatory reporting using blockchain technology [40]. The USA has no federal legislation, but some states are passing legislation enabling blockchain technology. Dubai and the UAE are leading the way with their robust data protection laws and classification standards [41].

Governments across the globe have taken different approaches to developing their blockchain ecosystems, with Malta creating a Digital Innovation Authority to enforce a blockchain technology approval process certifying platforms to enhance trust. The Korean government has adopted blockchain for identity management [42], and e-Estonia has its application in healthcare innovation [43]. The Caribbean has adopted blockchain technology in tourism, for the development of their economy [44]. A report by the Joint Research Centre (JRC) was submitted by the European Commission [30] which discusses a few use cases, including citizen's identity management, taxation reporting, development, and management, facilitating new decentralized business models without intermediaries, and e-voting, in which blockchain can play a major role in assisting governmental decisions. The implementations were divided and aimed at two governmental levels: national and local. Some of the proposed projects for blockchain-enabled service models as mentioned in the European Commission report [28] are presented in Table 1.

The Gulf state's positive ranking on eGovernment Readiness (EGDI) reflects major ongoing efforts that regional governments have made to improve the overall ICT infrastructure across ministries and government agencies. Among the Gulf Cooperation Council (GCC) countries, the United Arab Emirates (UAE) has the highest e-government development index, followed by Bahrain, Saudi Arabia, Kuwait, and Oman [45]. Among the worldwide top 40 countries leading in the e-government development, the UAE is grouped with countries with a very high online service index (OSI), at a rank of 21, jumping eight

positions up from the 2016 rankings [9,46]. Dubai—a significant emirate in the UAE—ranks at four (on a scale of ten) in terms of e-government participation initiatives and lies within a 'high' category in terms of e-government development [9].

In the Middle East, the Bahrain government's General Directorate of Traffic [47], as a part of the Smart Government initiative, has let out a tender to instigate the implementation of distributed ledger technology for monitoring and streamlining the vehicle registration process. The proposal mainly focuses on reducing cost, improving transparency and data integrity, and improving data management. The forecasted developments in the public services when explored show that blockchain can reduce bureaucracy, increase transparency and efficiency of administrative processes, and increase trust in the system [30].

**Table 1.** Proposed blockchain-based projects [28].

| Project Name | Country of Implementation | Field of Implementation | Level of Government Involved |
| --- | --- | --- | --- |
| Exonum land title registry | Georgia | Land title registry; property transactions | National |
| Blockcerts academic credentials | Malta | Academic certificates verification; personal documents storage and sharing | National |
| Chromaway property transactions | Sweden | Property transactions; transfer of land titles | National |
| uPort decentralised identity | Switzerland | Digital identity for proof of residency, eVoting, payments for bike rental and parking | Local (Municipality of Zug) |
| Infrachain governance framework | Luxemburg | Blockchain governance | National |
| Pension infrastructure | The Netherlands | Pension system management | National |
| Stadjerspas smart vouchers | The Netherlands | Benefit management for low-income residents | Local (Municipality of Groningen) |
| Fourth industrial revolution and smart customs | South Korea | Maritime Export Logistics; Import and Export clearance at customs department; Cross-Border Data Exchange | National and International |
| Blockchain-city-Melaka Straits City | Malaysia | Tourism | National |
| Tradelens | KSA | Supply chain and logistics | National |

## 3. Benefits and Challenges in Adoption of Blockchain by Governments

A blockchain is a database that has a digital ledger, otherwise called the smart contract that stores the details of all the transactions and information exchanged in blocks. It is a disruptive and decentralized technology that can be implemented to safeguard the rights of all the stakeholders on a digital platform, eliminating the need for a central administrator [48]. While a majority of researchers and practitioners argue about its abundant potential benefits and application areas in government, the literature also presents various challenges that need to be addressed.

In 2018, in a report on cryptocurrencies and blockchain in Europe and central Asia, the World Bank stated that "policymakers should strike a balance between curbing the hype surrounding these new technologies and exploring the essence of these new opportunities" [29]. The European Commission summarized the technical and legal challenges to the government's use of blockchain technology as lack of policy framework for cryptocurrencies, integration with existing systems, scalability issues, blockchain to blockchain interoperability, and enforceability of smart contracts [29]. The lack of infrastructure is also identified as one of the important obstacles to implementing this technology across the globe by governments. In this section, we review adoption challenges and how governments have faced them to encourage innovation, considering the benefits reaped out of this technology. Table 2 presents the benefits and challenges in the implementation of blockchain technology in various applications of government services.

**Table 2.** Benefits and challenges faced by governments in blockchain adoption.

| The Study Found in Literature/Use cases: Govt. Adoption of Blockchain | Benefits | Challenges |
|---|---|---|
| Adoption of blockchain for identity management, with a focus on the Korean Government [42] | Data integrity and Reliability; reduce the cost of service delivery; A user-centric personal data management without a central authority. Allows for quicker data access [42] | Educating the public sectors on the blockchain, privacy, and, regulatory concerns [42] |
| Key regulatory challenges of blockchain adoption in the EU and US. It discusses the hands-off approach initiated by both countries, and how this has leveraged the adoption of blockchain [49] | Promotes a clear understanding between cryptocurrency and blockchain [49] | Lack of adequate knowledge and blockchain expertise from regulators [49] |
| Automating the process of regulatory reporting using blockchain technology in the UK [40] | Reduces duplication, efficient regulatory reporting system; Creates a better understanding of blockchain for regulators by making them use the technology in regulatory reporting. | Educating regulators and other stakeholders, on confidentiality issues [40] |
| e-Estonia—Application of blockchain in healthcare innovation in Estonia [43] | Promotes better understanding of regulators, governments, and healthcare providers | Storage problems and end-user are responsible for data [43] |
| Use of blockchain to boost tourism among small economies [50]. | Commercial opportunities for small countries and improved stakeholder knowledge | Educating stakeholders on blockchain and regulatory gaps [50] |
| Adoption of blockchain technology in Tourism and the implications for tourism development in the Caribbean economy [41]. | Boosting tourism revenue; Launching the first digital legal tender in the Caribbean | Lack of IT infrastructure and government support for new technologies [41] |

### 3.1. Benefits

As already discussed, leading countries of the world and governments are utilizing the benefits provided by blockchain technology [47], and it has recently come to the forefront of research because of its potential benefits for many industries. Therefore, this study found it necessary to further probe the benefits of this technology (in addition to the ones discussed above).

Blockchain can reduce the overhead costs for small to mid-size businesses (SMBs) [48,51]. Saudi-based company Aramco has implemented a blockchain-based system for handling payments within peer-to-peer networks and has projected to bring about a cost efficiency of about 5%. Maurya [52] highlights the fact that both artificial intelligence (AI) and blockchain have contradictory properties, but when combined can lead to improved efficiency and enhanced security. The combination of these two technologies can lead to a faster and better decision-making process [52,53].

A blockchain is entitled to be completely reliable when it comes to safety and identity management [53,54]. The reliability of the available current data is important for governments to make any decision, however, the data collected from different sources may or may not be reliable at all definite time intervals. With decentralized data on blockchains, all the active nodes within the network would maintain a copy of the distributed ledger [55]. Hence, the reliability of the system is achieved and makes the network dependable for all nodes.

There is a necessity for the public—verifiable contracts with an increased level of transparency in data utility. Neisse et al. [56] developed a blockchain-based system that supported the properties of accountability and inception tracing, paving the way for publicly auditable contracts with privacy protection of individuals.

### 3.2. Challenges

The literature review reveals that several opportunities are available for utilizing blockchain in various government and private sectors; however, there are still some challenges to be addressed to achieve better utilization of this technology.

The topic of privacy in blockchain has been debated vigorously by practitioners and researchers. In the public blockchain system, the data on the network are made available to all authorized users [20,57–59]. This becomes a point of concern in safeguarding the privacy factors of individuals in the network and the data as a whole. Private blockchain

ledgers, on the other hand, would ensure restricted data access by authorized personnel [31]. Blockchain is much more secure when compared to central databases for safeguarding the data [57]. However, several security risks exist, which must be examined and analyzed before the actual data transaction takes place. The infamous Decentralized Autonomous Organization (DAO) attack [60] has shaken the cryptocurrency world, as the hacker managed to scrap around 3.6 M ether from the network, leaving the developers and the crypto world in shock. It is suggested in the blog [17] that peer review of smart contract code and independent testing of the code can avoid such incidences in the near future.

When it comes to implementing and adopting blockchain by organizations or government institutions, the major hurdle that remains seems to be the lack of standards and regulations [20,51,53,58,61,62]. As with any emerging technology, it takes a while to be accepted and adopted by individuals as well as a wide range of industries. As reported by Forbes [56], the regulation authorities lack complete knowledge of any technological advancement, which becomes the same case with blockchain as well. The launch of new services, applications, or products based on a decentralized blockchain, needs to be governed under regulatory acts which are lacking currently. Blockchain promises transparency and verifiability of all the events; however, the organizations need to build robust laws and industry standards to ensure smooth functioning of the overall process which will eventually enhance the trust that an organization will have in the blockchain network. Such an integrated approach will not only facilitate the financial functioning and transaction monitoring, but also will guide the companies' auditing process [20,63,64]. Laws and regulations need to be introduced to govern the smart contract information transaction and also to facilitate the interaction among system participants.

Blockchains are well-known for their collaborative and cooperative environment. Though it is decentralized, the success of the projects is heavily based on collaborative governance [53,61]. Governance is required by all organizations and projects to reach common agreeable terms and take high-level decisions. It is also to be emphasized that most of the existing organizations run on a centralized governance model and that decentralized models are yet to be developed [60,62,65,66]. Since blockchain is still evolving, governance, policies, compliances, and standards are yet to be designed and developed for decentralized systems.

Figure 2 presents the challenges and benefits of using blockchain technology in a nutshell. This study systematically reviewed relevant research to understand the challenges and benefits regarding blockchain adoption by governments, and then, through empirical study, we evaluate those benefits and obstacles for the Dubai government.

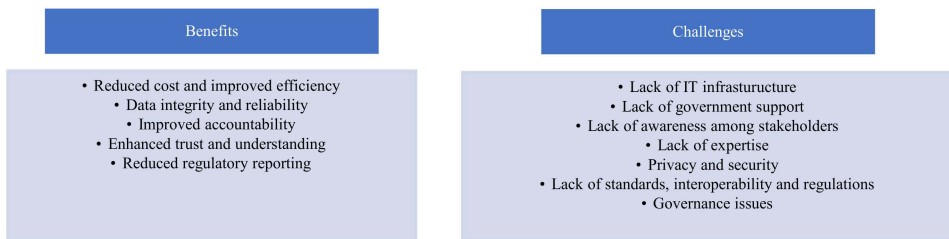

**Figure 2.** Benefits and challenges of blockchain for governments.

## 4. The Methodology

This paper studies the relevance of blockchain technology for digital governments. It explores the benefits and challenges that this technology is bringing to digital governments. This exploration is based on the extant literature, which is similar to the research conducted by other researchers, such as Bello et al. [67] and Elijah et al. [68]. In this regard, the study follows the Type V theory of design and action, exploring how the DED implemented its blockchain pilot project, and where the primary constructs are users, work context, information requirements, and the system architecture [69]. The benefits and challenges

that emerged through the process of researching the related fields using various sources being theoretical, called for validation via empirical research.

This study collected empirical evidence from the blockchain project, deployed in the DED, one of the seventy government departments in Dubai [70]. Given the specificity of the data sources used for this study, our methodological choice is a case study analysis. The case study method was chosen because of the limited research on emerging blockchain technology. Moreover, case study research is preferable when the research is mainly exploratory, rather than based on frequencies or incidence. The case study has been based upon the key informant interviews method. Six senior decision-makers and IT practitioners at the DED were used as the prime sources of the information. The interviewees were chosen for their specialized knowledge and experience with the use of blockchain technology in providing services. The qualitative data collected through semi-structured interviews are complemented with the information from desk research, and additional files are provided for details. Interview respondents were asked to validate the benefits and challenges identified through the literature review. The analysis proceeded according to the five steps outlined by LeCompte [71], namely tidying up, finding items, creating stable sets of items, creating patterns, and assembling structures. Constant comparison analysis was undertaken deductively to code responses into the pre-determined benefits and challenges [72].

Case studies can be used with any philosophical perspective of empirical research, be it positivist, interpretive or critical. Using a case study approach, the researchers explore the benefits and challenges based on the perspectives of the senior executives and IT practitioners. The positivist philosophy used for testing the theory is used to find the benefits and challenges validated by the same decision-makers and IT practitioners [73]. This led the researchers towards the use of positivist qualitative philosophy to validate the benefits and challenges of using blockchain implementations by various governments. Results of the analysis helped in proposing policy steps for governments to exploit the full potential of blockchain technology.

## 5. The Dubai Economic Department (DED)—Case Study Assessment Framework

There are fifteen government departments across Dubai offering nearly 1875 services, out of which 309 services are intra and interdependent. The DED is one of these government departments, which is responsible for developing economic plans and policies and provides services to domestic and international investors and businesses. The DED and its agencies develop, promote, and support foreign direct investment opportunities to facilitate investors. It also facilitates trade by creating an enabling environment for Dubai's exporters.

Dubai launched a blockchain strategy to deliver more seamless, safe, efficient, and impactful city experiences [74] and it plans to achieve efficiency by using blockchain in 100% of applicable government services. As per the recent reports [75], financial regulators in Dubai are working on crypto regulations and cryptocurrency laws. Dubai Ministry of Economy identified that crypto and asset tokenization was the fundamental support for doubling the size of the economy within the next decade [75]. Growing experimentation with blockchain technology and pilot projects that have already reached the production phase provides an opportunity to analyze the potential of blockchain technology, based on the empirical evidence. Accordingly, our study adopts an empirical approach to analyze the potential of blockchain in the public sector.

Researchers proposed a framework in a Joint Research Centre [30] report, to support the European policymaking process. This framework was developed for the generalization of collected data, and it consists of several elements covering institutional, functional, technical, and economic aspects of the case studies. Our study followed this framework to develop a similar case study assessment framework (Table 3), to provide evidence-based scientific support to our empirical findings.

**Table 3.** DED—Case study assessment framework.

| 1. Case Study General Features | | | | | |
|---|---|---|---|---|---|
| Level of government involved | Public services provided/enabled | Cross-border aspects | Cross-sector aspects | Location value creation | Openness of software |
| National | Issuing or modification of a commercial license to domestic and international investors and businesses | None | Other license issuing authorities, other government entities, and private entities, such as banks, telecom, real estate, and other financial institutions. | Across mainland and free zones | Propriety |

| 2. Functionalities | | | 3. Governance | | | 4. Usage | | | | |
|---|---|---|---|---|---|---|---|---|---|---|
| Institutions disintermediated | Functionalities provided | Roles Included | Blockchain governance architecture | Consortium governance | Current usage | capacity | Throughput | scalability | Maturity |
| Automation of internal approvals within the government entities. | – Issue, renew, modify, or cancellation of trade name<br>– Issue, renew, modify, or cancellation of trade license<br>– Issue, renew, modify, or cancellation of commerce registry<br>– Integration with other license issuing authorities<br>1. Integration with information consumer organizations<br>2. Integrated with Know Your Customer (KYC) registry | – Data Publishers: government license issuing authorities<br>– Data Subscribers: government, private<br>– Registry Service Provider | A hybrid permission-less governance model | Decentralized | Six to seven government entities transacting in this project | 400–500 tps | – Hyperledger supports 40 tps,<br>– CORDA supports up to 500 tps<br>– Hyperledger Besu | 500 tps | Production stage, following the successful execution of proof-of-concept |

| 5. Technical Architecture | | | | |
|---|---|---|---|---|
| Blockchain Interface | Corporate Registry Back-Office Portal | Blockchain peer node | CA Node | Orderer Node |
| Encapsulates all blockchain logic for Dubai Business Ledger. Encapsulates interfacing with its corresponding blockchain node. | Provide portal screens for the back-office operations of Corporate Registry, such check/audit exchange of trade license | Host Blockchain software and corresponding smart contracts of corporate registry | Authenticates all transactions | Broadcasts validated transactions for all organizations' peers. |
| **Underlying Software Technologies:**<br><br>– Hyperledger Fabric 1.1 SDK<br>– Node.JS<br><br>PostgreSql (RDBMS) | **Underlying software technologies:**<br><br>REACT (client-side development framework)<br>Node.JS | **Underlying software technologies:**<br><br>Hyperledger fabric Software V 1.1<br>CouchDB | **Underlying software technologies:**<br><br>Hyperledger fabricSoftware V 1.1 | **Underlying software technologies:**<br><br>Hyperledger Fabric Software v1.1 |

**Table 3.** *Cont.*

| 6. | Challenges | 7. | Benefits |
|---|---|---|---|
| | – Low utilization<br>– Technical immaturity<br>– Lack of skills<br>– Lack of awareness and trust<br>– Governance<br>– Migration challenges | | – Improved trust<br>– Standardization of regulatory checks<br>– Reduced operational costs<br>– Increased customer satisfaction<br>– Authenticity of documents<br>– Improved accountability<br>– Positive environmental impact |

### 5.1. Case Study General Features

The DED's e-government evolution started by having a website to be used for listing business activities. E-services allowed customers to search the activities provided by the DED as well as to find "terms and conditions". In the next stage, the DED enhanced its web services by providing two-way communication for instant license partner approval services. Next, the government of Dubai made a strategic decision in 2012 to achieve a transactional stage of e-government through the provision of high-quality customer-focused e-services (Dubai. ae). Following the directions, all fifteen departments of the Dubai government re-designed their governance, policies, and technology platforms to provide e-service to their customers.

During the transactional services stage of e-government, it required the DED take at least 7 days to issue a license for a commercial representative office. As shown in Figure 3, from an investor's perspective, this journey meant accessing various government department online services for taking approval, the contract development, registration, gaining utility connections, and then civil defense. It required the investors to repeat the business registration process at each of these departments. Moreover, each of these departments required similar documents, such as passport copies, ID copies, labor cards, etc., resulting in repetitions of documents. Furthermore, the DED lacked a unified business registry, across mainland and free zones, making it difficult to enforce common business rules, as required by law. It was possible to reserve duplicate tradenames across different entities due to the process in compliance. Modifications were not synchronized and service providers (such as financial institutions, government agencies, etc.) had to integrate with many different systems/authorities to complete their daily processes. Investors starting a new business were not only facing a long processing time of around three months to open a commercial bank account, but they were also unable to transfer goods from the free zones to mainland, without a local representative. Moreover, free zone firms were unauthorized to provide services to the mainland. Siloed, disorganized and paper-based practices negatively affect customer experience. For financial institutes, such as banks, there was a high operational cost involved in collecting investors' data and their continuous updates. Moreover, there was difficulty in complying with regulatory requirements, leading to penalties in cases of non-compliance, e.g., money laundering, fraud, and terrorist financing. This resulted in low customer satisfaction with banks due to lengthy procedures. In addition, non-unified standards and lack of governance on business licenses from different authorities were other challenges faced by licensing authorities. It was realized that these existing governance structures, laws, and models were not enough to achieve the goal of creating a global business hub and therefore it required significant changes.

Shift to e-services simplified the access to government-related services to people and businesses, and considerably reduced the number of people visiting the counters. However, while each government department on its own was performing very well by providing excellent e-services to its customers, it was realized that the lack of integration and lack of information sharing among these departments was still arduous for customers when it came to be dealing with various departments for any given business activity. It was still causing customers to face long and complicated processes and duplications in the requests for documents by each department.

In a constant endeavor for improvement, the Dubai government launched an initiative called 'Connected Dubai' in 2015, a framework for integrating and enhancing services across government entities. The key focus areas of this framework were on redesigning services and enabling information sharing across the organizational boundaries. Apart from revamping governance structures and policies, technology was used as a building block towards the smart digital transformation processes at the government level. In support of the government level 'Connected Dubai' initiative, in 2017 the DED revamped its strategy and services and came up with blockchain use cases to improve the business environment and accelerate productivity growth. Figure 4 shows the blockchain roadmap followed by the DED.

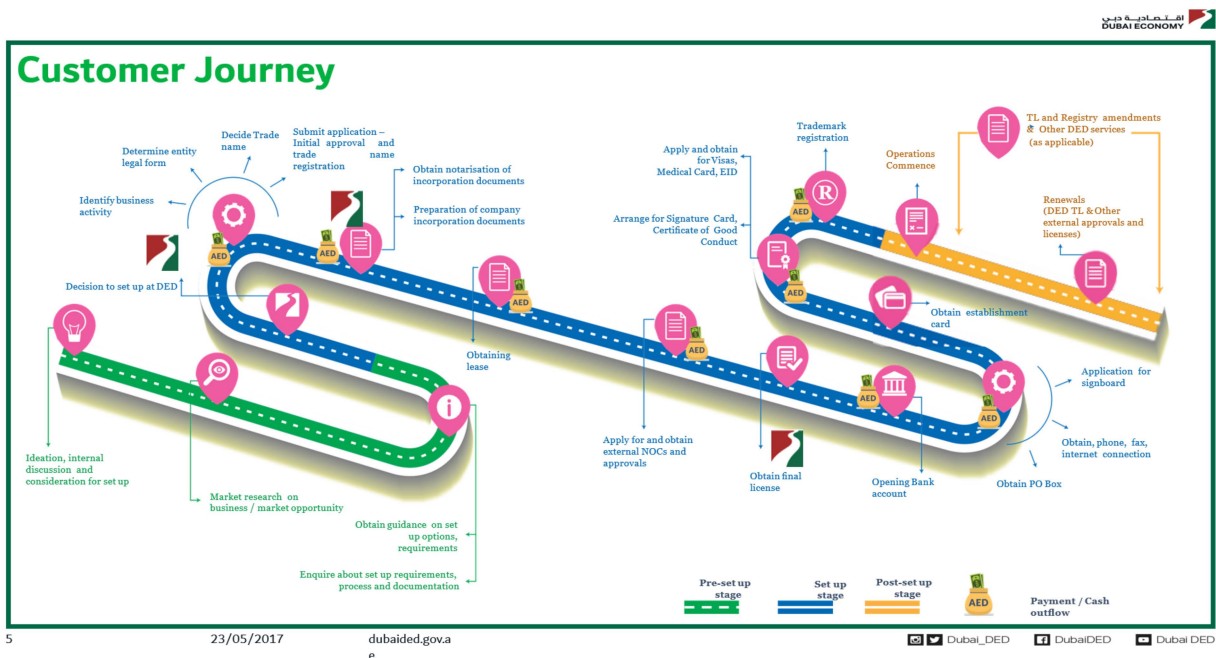

**Figure 3.** Customer Journey before blockchain implementation ("Source: Dubai Economic Development").

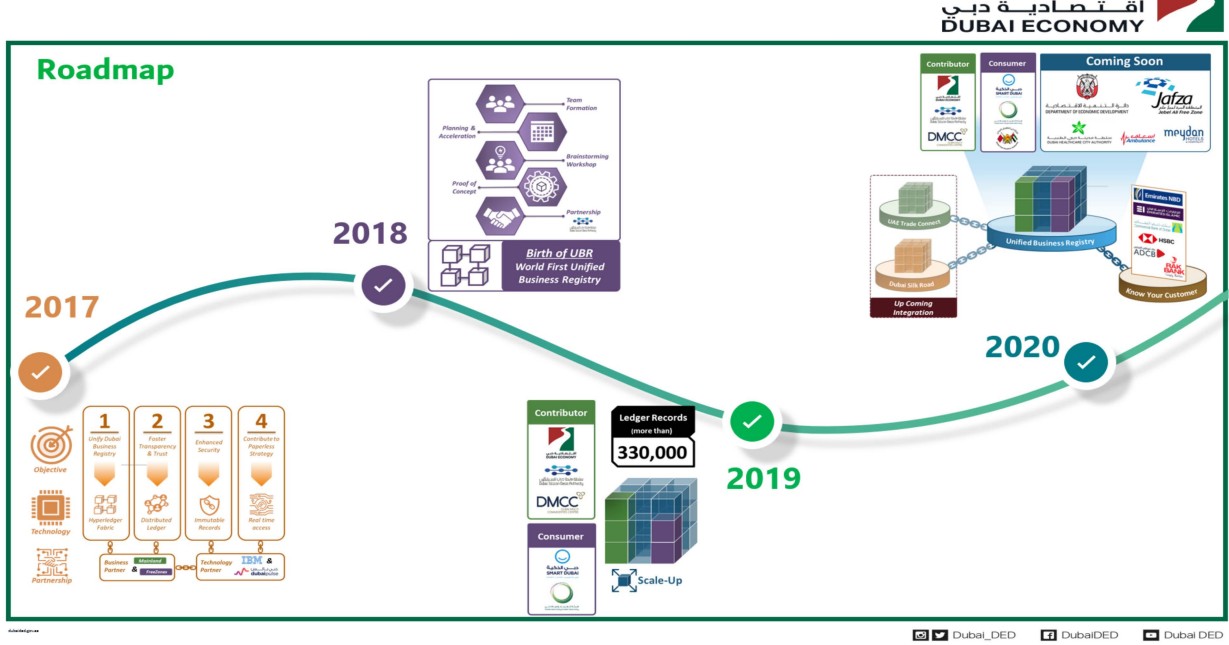

**Figure 4.** Blockchain roadmap followed by the DED ("Source: Dubai Economic Development").

### *5.2. Functionalities*

In 2018 the Unified Business Registry (UBR) was developed to provide common infrastructure and services for information sharing, unified data standards, and data registries. It is a government-wide data management service to enable authorized staff (or users) access to share departments' data sources. Services were placed on UBR to offer innovative customer-centric services and service packages and a seamless experience for users. Using this system, customers can apply for online services from one place and necessary documents are requested only once.

UBR is integrated with trade name registration, commerce registration, trade license registration, and other business activity entities. An issuing authority pushes trade name

information to the blockchain whenever a trade name is issued, renewed, modified, or canceled. Likewise, the commerce registry issuing authority pushes commerce registry information to the blockchain whenever a commerce registry is issued, renewed, modified, or canceled. Similarly, the trade license registration issuing authority pushes trade license information to the blockchain whenever a trade license is issued, renewed, modified, or canceled. The DED pushes business activity information to the blockchain whenever a business activity is registered or modified. Mapping has been added for activity codes used by other licensing authorities. In 2019, UBR scaled up to integrate with other license issuing authorities, such as Dubai Multi Commodities Centre (DMCC) and information consumer organizations such as Dubai Electricity and Water Authority (DEWA), which need business consumer's data coming from free zones and mainland entities. All internal approvals within the government are automated through integration and information sharing. Currently, in phase 2 termed 'Innovation Beyond Borders', UBR has been integrated with the Know Your Customer (KYC) registry which has entities, such as banks (eight banks) and other financial institutes, on board (Figure 5). Currently, while UBR is in the live stage, the KYC registry is in the final testing stage [75]. Customers are presented with add-on services from private sectors (banking, telecom, real estate) all within the same application. The investor's required data and documents are now placed on the UBR, accessible to other government entities. Registries are formed on people's basic information and business entities' information related to each department. For a user, access to this registry is through 'MyID'—a single identity across services. Using this single access, users are identified and able to use any of the government services on this registry.

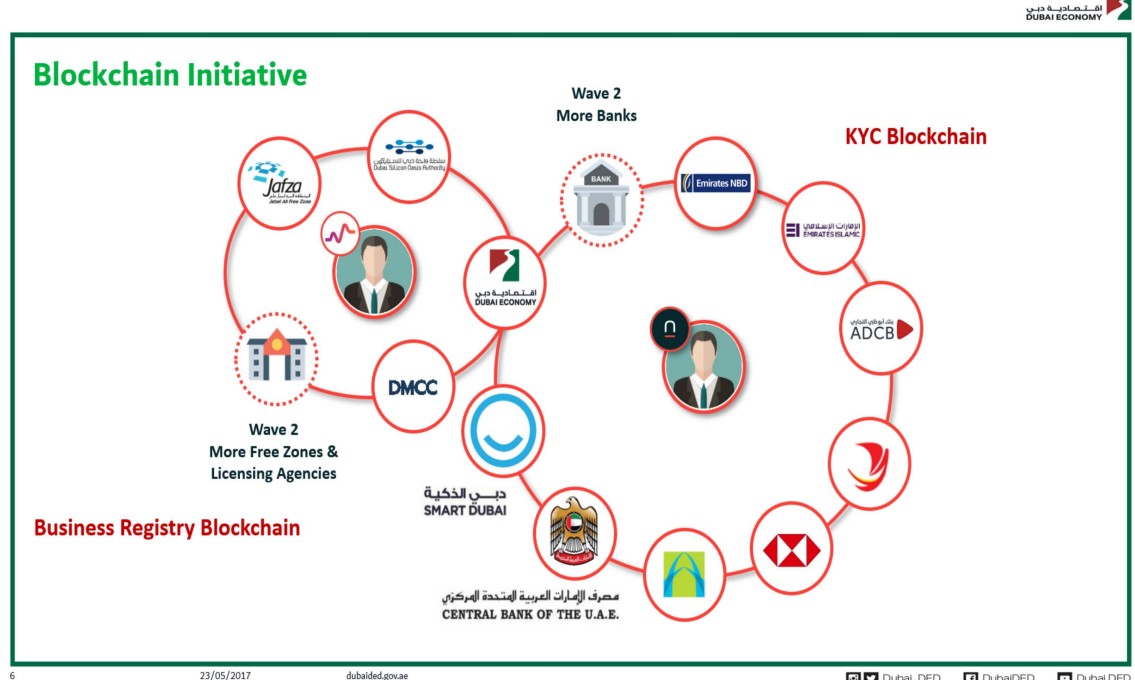

**Figure 5.** Blockchain initiatives at the DED ("Source: Dubai Economic Development").

In the future, the DED plans to integrate its blockchains with other government registries for sharing of information across various industries and sectors. The DED also intends to bring all other free zones, government entities, insurance companies, healthcare, and hotel industries on board, for the exchange of real-time data required for legitimate business processes across the industries. They also have plans to engage other GCC countries to be part of this bigger, wider blockchain technology infrastructure [70].

*5.3. Governance*

In terms of governance, DED blockchains follow a private permission-less blockchain governance architecture. The DED controls the consortium and allows others to join based on their permission. The DED utilizes the controlled governance structure as the project is mainly controlled and directed by the DED; however, as more government entities are added, it will be floated across nodes. The DED holds the right to control what data can be read by other government entities, what can be written on this ledger, and what information can be visible to others.

UBR Roles

The DED has identified three classes of the participants in the unified business registry as data publishers, data subscribers, and registry service providers.

Data Publishers—the entities that publish data in the registry. In the initial phases, the only license issuing authorities will be the producers of UBR data as they will be publishing an index of their issued trade licenses. They will also be publishing the private data of licenses using private ledgers, set up between the issuers and subscribers. While the DED registers companies on the mainland, DMCC and Dubai Silicon Office (DSO) perform the same in free zones. DMCC and DSO are data publishing nodes providing company registry data to this URB to be consumed and shared with everyone part of this chain.

Data Subscribers—the entities that subscribe to data from the UBR registry for their business transactions. These entities could be government, private, and license issuing authorities as well. These entities will be classified into the following different categories of the subscription:

- All trade license entities that will have access to trade licenses from issuing authorities within the emirate, e.g., Dubai Silicon Authority, Dubai Chamber of Commerce, Ministry of Labor, Dubai Courts, Dubai Statistics, etc.
- Access based on business activities—entities that need access to trade licenses from all issuing authorities but only for specific types of activities, such as Dubai electricity and water authority (DEWA), and Etisalat (a telecommunication company).
- Access based on license type—entities that need access to trade licenses based on license type, such as Dubai Customs and Dubai Port Authority, need access to Trade/Commercial Licenses.
- Access based on a need-to-know basis—entities that need access to selected trade licenses granted by the license owner, e.g., banks, other private commercial entities, and free zones.

Registry Service Provider—entities that manage UBR indexes, regulate and monetize all transactions (current and future) in the registry. The entities will provide UBR services to private entities, such as banks.

*5.4. Usage*

Currently, six to seven government entities are transacting in this project. TPS is calculated based on the use cases, the workload, and the transaction between every organization. The throughput of the Dubai Pulse platforms can serve thousands of users. Hyperledger Fabric, right now the existing use case, is running on the 40 tps, the CORDA supports up to 500 tps. The platform is capable of handling 400–500 transactions per second, depending on whether it is Hyperledger fabric, Hyperledger Besu, or CORDA blockchain protocols.

*5.5. Technical Architecture*

The DED blockchain has been set up as private permission-less blockchain governance architecture. With these initial participants, any new node or new entity to be added needs acceptance or a vote of approval from the previous entities. It is designed in such a way that a new entity or node does not need every other node's approval, but can have automatic approval. To gain automatic approval, a new entity or node needs at least three members to approve to be a member of the node. The DED can control who has access to which data

from the smart contract itself by adding parameters. The architecture built is promising as no direct access is given to the back-end of the node to avoid any problems of data or business breaches for the business or consortium owner. A striking feature provided by the DED to the miner nodes is that the participants can see the smart contract that is installed on their node, permissions granted, but not the business logic.

In phase 1 of implementation, the DED was connected to government entities, such as Dubai electricity and water authorities (DEWA), Dubai Silicon Oasis Authority (DSOA), and Etisalat (a telecommunication company). Once the business trade licenses are on the DED blockchain, customers do not need to visit or deposit documents at the various department for electricity and water or telecommunication connections, for their businesses. In phase 2, all other economic departments in the UAE are connected to the same ledger across free zones and mainland in the UAE, to have a unified trade license, following the same regulations, thereby eliminating any frauds or customer issues.

The DED emphasizes using Golang, a NodeJS programming language, to write smart contracts, and leverage solutions for blockchain platform-as-a-service. Multiple-choice protocols, such as Corda, platforms such as Ethereum, and Hyperledger, with many more, are made available and supported in the development stage. Along with this, the DED has monitored concerns, such as smart contract vulnerabilities, that might occur during programming stages. Blockchain services at the DED have built-in antivirus and anti-malware components where smart contracts are rescanned before installing them, first on the channel, then on the node of every organization. This is a groundbreaking effort, as it will not hold the malicious code in the smart contract and makes sure that the smart contract is secure before installing it to the node itself. In addition, direct access to the back-end of the business owner's node is denied ensuring complete safety for the data stored.

The model proposed is open to other government entities beyond Dubai, which offers them flexibility on the type of protocol they might want to choose. The DED is not strictly associated with any specific protocol but offers a flexible and open approach, and has multiple protocols, such as Corda, Ethereum, and Hyperledger. The number of protocols is increasing, and the DED emphasizes the fact that this will open a fleet for government entities. A use case is already built so that they do not have to rebuild or reinvest to match the platform. The platform is flexible enough, to understand all the use cases from various blockchain protocols and integrate them. Infrastructure is provided by the DED for the consensus algorithm and it is performed in-house along with the partner. A consensus algorithm will cater to all the various protocols that are there in the market. As of now, the DED has Corda, HyperLedger, and Ethereum as the three most commonly used protocols for the government entity. Once this is completed, the DED has the infrastructure set up and running and ready to approach different government entities, the Dubai economy being the first success story. A POC was built, and all the trade licenses were transferred to the blockchain and authenticated. Initially, this was deployed in the Dubai pulse blockchain and now they have five government entities connected to this node. This will eliminate the need to request a trade license from customers. Once they go to the blockchain, customers can show the blockchain code, and the government entities can verify it automatically.

Dubai pulse, a high-quality verified open dataset, which is legally recognized as a reference register, is a shared resource for government entities that provides several cloud-based services based on usage or pay-as-you-go basis, for each government entity, thereby facilitating the decision-making capabilities of the DED. The platform compiles all government data in one place, where the right information can be provided to the right people whenever they should need it. 'Dubai Pulse' will enhance operational efficiency by reducing data access costs. It will facilitate the exchange of open and common data between the public and private institutions and individuals, which will eventually contribute to the complete digital transformation of the Emirate of Dubai. The DED is providing connected services to customers on the unified business registries (UBR) blockchain of Dubai pulse. Such a process will also allow customers to access the services associated with it instantly rather than waiting for many days. From a technical perspective, this new platform

provides access, for the first time, to live and up-to-date data about the city. It includes three layers of data: "the first will be free of charge and caters to the public, while the second offers a thorough analysis of the data (in exchange for a fees) to be used for academic, professional, commercial, and economic purposes. The third and the final layer includes data accessible exclusively to Dubai Government entities".

### *5.6. Benefits and Challenges*

A review of benefits achieved by the DED, by using blockchain technology to streamline its processes include costs savings, customer satisfaction, time and efficiency savings, reliability, environmental benefits, enhanced trust, and improved accountability. These are all discussed in detail in the next section (Section 6.1).

A review of challenges faced by the DED, by using blockchain technology to streamline its processes include low utilization, technical immaturity, lack of skills, lack of awareness and trust, governance issues, and migration challenges. These are all discussed in detail in the next section (Section 6.2).

## 6. Insights from the Case Study

The DED has employed blockchain-based applications to improve the digital services provided to citizens and expats [70]. Empirical intervention shows that the DED has matured from the initial emerging services stage of e-government evolution to the advanced stage of providing connected services to the public. After adopting the blockchain platform the DED is expecting to further attract businesses into the region and increase the ease of conducting business rankings of Dubai and UAE when compared to other countries [75,76].

### *6.1. Benefits*

The benefits achieved by the DED by using blockchain technology to streamline its processes are summarized in Figure 6 below:

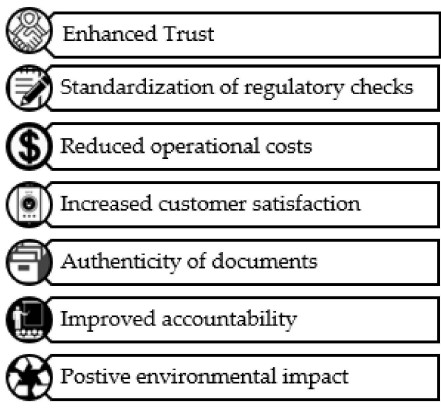

**Figure 6.** Benefits achieved by DED.

- Enhanced trust: The DED benefited by having simpler and faster processes across different authorities. It helped in creating trust among investors as any modification made to the investor's data can be synchronized across different authorities automatically and instantaneously. The surface area of integration required by the service provider is vastly reduced. Various policies and rules are in place to secure the access and authority levels of the customers. These measures help in increasing trust among participants.
- Standardization of regulatory checks: By standardizing all the regulatory checks, by unifying and verifying all the processes on the blockchain, resulting in drastic time savings in various business processes.
- Reduced operational costs: High operational costs reduction is achieved by using blockchain-as-a-service based on a pay-as-you-go model with different subscription

packages. The DED is achieving around 85% cost savings by providing connected services to customers on the unified business registries (UBR). Cost reductions have been possible as they do not need to build the entire infrastructure by themselves, and they do not need to hire subject matter experts in this domain to maintain and develop the use case. Sharing information across entities in a common registry resulted in the reduction of the overall cost of information management, improvement in data quality, and creation of opportunities for collaboration.

- Increased customer satisfaction: Re-use of information from trusted authorities (create once use many) increased customer satisfaction by relieving constituents from the burden of submitting copies of documents issued by government entities. Moreover, real-time data sharing between various entities resulted in efficient, trusted, and secure services, thus leading to intangible benefits in terms of customer satisfaction and trust.
- Authenticity of documents: By allowing data in the blockchain, it brought authenticity to the documents submitted, as otherwise documents submitted could have been tampered with. Security of data is ensured through various policies and controls in place to secure the access and authority levels of the customers.
- Improved accountability: By choosing to use blockchain instead of simpler solutions based on database systems, the DED has given precedence to secure systems that are cost-justified as well. Accountability has improved by using decentralized blockchain technology, which has created a link between other free zones and another mainland in Dubai.
- Positive environmental impact: It brings a positive impact on the environment by reducing fuel emissions. This is possible because of the paperless and travel-free transactions using this technology.

*6.2. Challenges*

Challenges obtained through the literature review were compared with ones faced by the DED. While there were some similarities, blockchain being the national strategy, the DED did not face challenges related to lack of IT infrastructure or lack of government support. A few additional challenges which surfaced through the empirical study are related to migration and underutilization issues. While the DED has achieved many benefits by using blockchain technology, the challenges faced them are summarized in Figure 7.

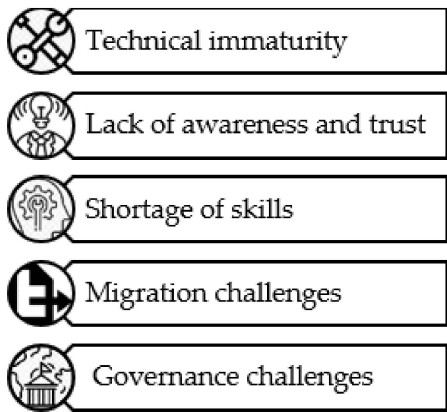

**Figure 7.** Challenges faced by DED in using the Blockchain strategy.

- Underutilization: Currently there are only a few participants in this chain. Data are lying there on the blockchain underutilized. The DED is finding it difficult to bring together required stakeholders.
- Technical immaturity: A consent mechanism is used for sharing data with other entities. However, it works on a trust basis, as audit rights or sharing of audit reports with partners is still in its formative phase.

- Lack of awareness and trust: To increase trust and awareness among organizations, the 'Dubai blockchain center' has been established for educating private and public organizations about the use cases of blockchain technology. It is helping the government to build and enhance strategies and regulations to help businesses to accelerate the adoption of blockchain. However, awareness among stakeholders is still low and organizations are skeptical about the use of blockchain technology.
- Shortage of skills: For blockchain, various skill sets are needed for front-end as well as for back-end development. Various required skill sets include chain coding and integration skills. Other skill sets are needed for the development of the protocol and the consensus, involving the back-end side. For the front end, there are skills available in the market to write smart contracts and leverage kind of required solutions for blockchain platforms as a service. However, there is a shortage of people with skills in back-end programming. To fill this gap, the UAE has set up a 'Dubai blockchain center' to offer training to people, offering certification courses to develop front-end and back-end applications on various blockchain protocols and use cases.
- Migration challenges: Organizations find it challenging to map legacy data into the new system. Once legacy data are mapped into the new system, they become incumbent to perform lots of quality checks because not all the data fetched from the legacy system will be of equal quality as needed in the new system. Thus, it is not only linking/fetching, but it also requires quality checking of the data.
- Governance challenges: Another challenge is in terms of changes in the governance architecture. Changing the centralized to decentralized governance model is a gradual process, however, it starts with people having trust in the platform.

## 7. Conclusions and Future Research

The ongoing exploration of blockchain technology by governments is growing exponentially. Several governments across the world make use of blockchain technology to improve their services in property transactions, land registry, identity management, and benefits management, to mention a few. Implementing blockchain technologies by many government agencies across the world witnessed an increase in transparency of transactions, improved and reduced redundancy, and an increase in data integrity.

This study explores the transformative and disruptive innovations portrayed by this technology. The exploration of the literature related to the use of blockchain technology by governments around the world helped in understanding what governments are currently undertaking with this technology. Furthermore, this exploration also helped in understanding the benefits and challenges faced by various governments in implementing blockchain to offer e-services. This report studied the transition of a Dubai government entity from a traditional to a blockchain platform. To provide evidence-based scientific support, this study developed a case study assessment framework, based on empirical findings. In-depth interviews and desk research helped in validating the benefits and challenges of blockchain technology, from a regional perspective. This empirical intervention helped in proposing the below policy actions that should benefit all blockchain-based pilot deployments.

### 7.1. Recommended Policy Actions

- To fully utilize the transformative power of blockchain, below policy actions by governments will spur the exploitation of the full potential of blockchain technology.
- Blockchain is a consortium and there have to be multiple participants in this. The public, as well as the private entities, need to increase the utilization of blockchain technology by plugging into this consortium to consume the data provided on the blockchain. The higher the utilization and integration, the better the benefits of this technology will be. Technology maturity will also increase with an increased number of participants on the blockchain platform.
- Moving to the future requires organizations to be well prepared to have new structures, systems, and securities. Migration from legacy to the new model requires an update

in infrastructure, linking of the two systems, fetching and mapping the data, and performing the quality checks on the mapped data.

- Organizations face a challenge in terms of changes in the governance architecture. Organizations need to understand that changing the centralized to decentralized governance model is not going to happen overnight. It is a gradual process that starts with stakeholders having trust in the platform.
- Disseminating awareness among organizations needs to be increased to gain participants' trust and for an increased participation.
- To overcome the shortage of skills, universities need to be advised to prepare graduates with back-end programming skills in Java, node.JS, Kotlin, Go, C++, C#, etc., required for building the protocol or automating onboarding of new entities in the blockchain.

### 7.2. Future Research

The case study presented in this paper represents the state-of-art development of blockchain technology in the public sector within the UAE. There is a lack of empirical evidence as to all analyzed services, including operational implementations, which are currently limited in scope to a single local or national administration. Future studies will focus on empirical evidence from a group of ongoing projects in the UAE, which has utilized blockchain technology for developing end-user services relevant to the public sector. A customized assessment framework will be developed to facilitate the collection of field data, a comparative analysis of case studies, and the generalizability of results. In addition, horizontal analysis of the deployments will help in refining policy actions that are required to support the development of this technology in the region.

**Author Contributions:** S.K.—conceptualization, methodology interviewed respondents, writing—manuscript draft preparation, investigation, data collection, visualization; second author: M.S.—conceptualization, validation; M.M.—conceptualization, methodology, supervision, validation, visualization; N.N.—literature review, blockchain, and digital governments, challenges, interview transcription, writing—manuscript draft preparation; M.N.—conceptualization, and validation. All authors have read and agreed to the published version of the manuscript.

**Funding:** Article processing fees will be shared by the University of Windsor, Canada and Zayed University, UAE.

**Institutional Review Board Statement:** The study was approved by Research Ethics Committee (REC), Zayed University, United Arab Emirates. Ethics Application Number: ZU20_048_F, dated: 3 March 2020.

**Informed Consent Statement:** Informed consent was obtained from all subjects involved in the study.

**Acknowledgments:** We would like to extend our sincere gratitude and appreciation to the experts at the Dubai Economic Department (DED) and Dubai Smart Cities for the technical insights, support and validation of this work.

**Conflicts of Interest:** The authors declare no conflict of interest and they have no known competing financial interests or personal relationships that could have appeared to influence the work reported in this paper.

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
