# Peer review of "Blockchain for Governments: The Case of the Dubai Government"

_sustainability, doi:10.3390/su14116576_

Round 1
Reviewer 1 Report
The author has written a well researched case study of dubai implementing blockchain for governmental purpose. It can be a beneficial startup for the government and researchers of same discipline.
The paper require extensive revamp of introduction that should contain only important information while removing the unnecessary details of blockchain benefits
The section of blockchain issues and challenged must be reduced to necessary details related to issues in governmental process. It is required to give a comparison how identified challenges are overcome by selected case study.
It is suggested to propose a model that can help governments in implementing successful blockchain in their services. The framework may require a validation based on expert opinion that were selected for case study
Author Response
Dear Reviewer,
We authors would like to thank you for your thoughtful comments and efforts towards improving our manuscript. The comments are very valuable and helpful for revising and improving our paper, as well as acting as significant guidance to our research.
We answered all the reviewers’ comments and modified the manuscript accordingly.

Reviewer 2 Report
The article presents the idea of an e-government based on blockchain. The main argument derives from the transparency and immutability of blockchain, allowing for the secure and open transmission and storage of data. Thus, the information related to the public sector is open to the public eye and protected from any kind of corruption or mischievous activities. Also, it manages to increase the speed of transactions in day-to-day operations within the government. The case of Dubai (UAE) is presented, where this kind of innovative technology has already been implemented showing great promise for future endeavors.
The tackled subject is something novel, an innovation in terms of data transmission and accessibility in regards to the government and public sector. The article presents the use case of Dubai and the benefits and challenges discovered after the fact. Tables are used for presenting the stages of development, the main architecture and the future progress envisioned by the authors. The usage of real data and the concrete evidence presented in the article that stems from the project manages to have, in my opinion, sufficient impact to prove the untapped potential of blockchain in the evolution of e-governments.
It is a thorough and well researched paper, making use of various tables and concrete evidence gathered from the chosen use case in order to transmit the main scope of the presented project. It is a quite extensive read but not too complicated or bloated. It gets the point across in a somewhat lightweight manner. I would recommend changing the somewhat old references and adding higher quality and more condensed tables in order to make the information more digestible.
In my perception, the article is well structured, explicit, thus giving any reader the opportunity to understand the subject matter. Also, the graphical representations are really interesting and help a lot in understanding the concepts, even if the reader is not familiar with software technologies.
The article addresses a topic of current interest: Blockchain technology implemented in the government systems of various countries, based on direct citizen-government interaction.
The conclusion has an atypical structure, being another section with sub-sections rather than a summary of the ideas debated in the article.
The references are mainly up to date except some that are as old as 2008, 2006 and 2000. More references regarding related work of interoperability of distributed ledger technologies should be provided and the rationale for choosing a particular blockchain (Hyperledger, CORDA, etc.) should be discussed, for example:
- Kassen, Maxat. "Blockchain and e-government innovation: Automation of public information processes." Information Systems 103 (2022): 101862.
- Carmen, Nadrag, et al. "Comparative analysis of distributed ledger technologies." 2018 Global Wireless Summit (GWS). IEEE, 2018.
- Assiri, Haitham, Majed Eljazzar, and Priyadarsi Nanda. "Blockchain in Saudi e-Government: A Systematic Literature Review." International Journal of Electrical and Computer Engineering 16.1 (2022): 11-19.
Following analysis of the text, several grammatical errors can be identified:
- Sector -> the singular noun sector follows a number other than one. Consider changing the noun to the plural form -> sectors
- In United -> the article usage may be incorrect -> in the United
- For -> it seems that preposition use may be incorrect-> to
- Public-> the noun phrase public sector seems to be missing a determiner before it. Consider adding an article -> the public
- Work loads -> the word work loads seems to be miswritten -> workloads
- Issue -> it seems that issue may not agree in number with other words in this phrase -> issues
- Policy making -> the word seems to be miswritten -> policymaking
- Customer related -> it seems that customer related is missing a hyphen. Consider adding the hyphen -> customer-related
- In order to -> the phrase in order to may be wordy. Consider changing the wording -> to
- Republic-> the article usage with the geographical name Republic of Moldova may be incorrect-> the Republic
- In -> it seems that preposition use may be incorrect -> (contributed) to
- Markets, -> the comma may be separating the subject and verb in your sentence. Consider removing it
- Block – the noun phrase block seems to be missing a determiner before it. Consider adding an article -> the block/ a block
- Need -> it seems that the verb need does not agree with the subject -> needs
- Crypto currencies -> it seems to be miswritten, consider replacing it -> cryptocurrencies
- Blockchain -> it seems to be missing a determiner before it, consider adding an article-> the blockchain
- Few-> it appears that the phrase only does not contain the correct article usage-> a few
- Arab -> it seems that there is an article usage problem -> the Arab
- (begin) with -> it seems that preposition use may be incorrect -> (begin) by
- Methodology -> the noun phrase seems to be missing a determiner before it. Consider adding an article -> the methodology
- There are multitude -> the plural form of be are does not seem to agree with the singular subject multitude. Consider changing the verb form -> there is
- With respect to -> the phrase may be wordy, consider changing the wording -> concerning for to
- Multitude -> the noun phrase seems to be missing a determiner before it. Consider adding an article-> a multitude
- Error -> it seems that error may not agree in number with other words in this phrase -> errors
- The emerging -> it seems that there is an article usage problem -> emerging
- Decision makers -> it seems that customer related is missing a hyphen. Consider adding the hyphen -> decision-makers
- Key-informants -> the word doesn’t seem to fit this context, consider replacing it with a different one
- Calls-> the verb calls does not seem to agree with the subject, consider changing the verb form -> call
- Theoretical -> the noun phrase seems to be missing a determiner before it. Consider adding an article-> the theoretical
- Nationally comparable -> it appears that nationally comparable is missing a hyphen-> nationally-comparable
- Basically -> it appears that basically may be unnecessary in this sentence, consider removing it.
- Stake holders -> word miswritten -> stakeholders
- Majority-> the noun phrase seems to be missing a determiner before it. Consider adding an article-> the majority
- Present -> the verb does not seem to agree with the subject, consider changing the verb form -> presents
- Industry wide -> it seems industry wide that is missing a hyphen. Consider adding the hyphen -> industry - wide
- Different -> it seems that preposition use may be incorrect -> in different
- Cost -> it seems that cost may not agree in number with other words in this phrase -> costs
- Saudi based -> it seems industry wide that is missing a hyphen. Consider adding the hyphen -> Saudi-based
- Blockchain -> it seems that there is an article usage problem -> a blockchain
- Peer to peer -> it seems peer to peer that is missing a hyphen. Consider adding the hyphen -> peer-to-peer
- Fee-> it seems that fee may not agree in number with other words in this phrase-> fees
- The reliability on -> incorrect preposition-> The reliability of
- Make – it appears that your sentence or clause uses an incorrect form of the verb make, consider changing it-> making
- On the system -> it seems that there is an article usage problem-> system
- Happening -> the word doesn’t seem to fit this context -> happenings
- Increased-> the noun phrase seems to be missing a determiner before it. Consider adding an article ->an increased
- Literature -> it seems that there is an article usage problem -> the literature
- Collection-> the noun phrase seems to be missing a determiner before it. Consider adding an article ->the collection
- Data on stake-> incorrect preposition -> data at stake
- Privacy-> the noun phrase privacy breach seems to be missing a determiner before it. Consider adding an article -> a privacy/the privacy
- Critical -> the noun phrase critical infrastructure seems to be missing a determiner before it. Consider adding an article ->the critical
- Experiment -> -> the noun phrase seems to be missing a determiner before it. Consider adding an article -> an experiment
- Public-> -> the noun phrase seems to be missing a determiner before it. Consider adding an article ->the public
- commission -> it appears that the word commission may be a proper noun in this context, consider capitalizing the word -> Commission
- blockchain enabled -> it seems that is missing a hyphen. Consider adding the hyphen -> blockchain-enabled
- final -> there is an article usage problem ->the final
- live -> there is an article usage problem -> the live
- future -> there is an article usage problem-> the future
- free-zones -> the word doesn’t seem to fit this context, consider replacing it with a different one -> free zones
- Dubai-> there is an article usage problem-> the Dubai
- On-going -> ongoing
- Further-> a missing comma after the introductory phrase Further-> Further,
- Being -> the verb being appears to be unnecessary here
- Each and every -> may be redundant
- Travel free -> it seems that is missing a hyphen. Consider adding the hyphen -> travel-free
- In order to -> the phrase may be wordy, consider changing the wording-> to
Author Response
We are grateful to the editors and peer reviewers for their thoughtful comments that helped us improve the quality of our
manuscript.
We answered all the reviewers’ comments and modified the manuscript accordingly.
Please see the attachment.

Reviewer 3 Report
In the abstract, the authors state that they have conducted an analysis of the public sector impact of blockchain technology and its implementations for various governments. However, it is believed that the analysis is not very relevant due to the small amount of case studies analyzed. The case study on the evolution of e-government for the government entity of Dubai in the United Arab Emirates (UAE) is interesting, but the analysis does not contain a sufficient description of the infrastructures built over the years, so that any reader can fully understand the process of technological transformation and be able to replicate the results obtained. The analysis is superficial and does not highlight the technical progress actually achieved, but provides a summary assessment of the benefits obtained and the challenges still unresolved. The title is captivating: a “Blockchain” for governments, but the results described in the paper do not fully describe any infrastructure. Making a Blockchain today is a real challenge where the benefits of immutability, security, decentralization, interoperability, incorruptibility and even efficiency are currently impossible to obtain in a single infrastructure. With this premise, analysis and literature reviews must be structured in a much more precise and exhaustive way, indicating the real benefits that can be obtained on the basis of the architectural choices and policies implemented, but above all, they must be reformulated by providing the reader with the necessary technical elements: to evaluate the results described in the use case and to refute the results obtained on the basis of empirical evidence. The paper, on the other hand, seems to generally describe the "potential" benefits that can be obtained from the adoption of Blockchain technology within e-government processes.
In the paper
The numbering of the references in the entire paper is incorrect, that is, it is not indicated in the paper in an increasing sense and based on the citation order adopted (already on the first line 29 there is a reference [37] ??? and not [2]).
Abstract
Review the form and English from line 18 to line 20 (not only… also).
- Introduction
Line 31: “ … the concept of e-government to enhance the quality of life… ”. In this historical period due to the Covid-19 pandemic, e-government processes have become a necessity, rather than an ambition to improve the quality of life. Authors are requested to underline this importance and also to indicate bibliographical references.
Line 40: “… other bureaucracy issue for government employees… ”. It is not clear what this bureaucratic issue is, please the readers to specify this statement better.
Line 40: “… Some governments… ” Which governments? This aspect is crucial for the analysis conducted by the authors.
Lines 84-92. The paper contains 4 questions, which have also been duplicated several times within this paper, but which do not facilitate the reader either to read or understand the content of the paper. Please eliminate them and avoid repetitions.
- Background
Line 115-116 “ Several other attacks such as ransomware and DOS attacks dominated the past era ...". This claim is not supported by scientific relevance. These attacks are still dominant. Insert bibliographic references.
"3. Methodology " and "4. Literature review "
IMPORTANT!!! Authors are strongly advised to reformulate paragraphs 3 and 4 entirely, contextualizing the adoption of blockchain technology in the e-government process and describing in detail the state of the art achieved by the use of this technology and the new and innovative computerization processes and dematerialization actually achieved to date, without generalizing by describing the set of challenges and benefits achieved by the Blockchain.
- E-Government Evolution Internships
This paragraph should illustrate to the reader the evolution of e-government processes and the existing literature for the Dubai government entity in the United Arab Emirates (as stated by the authors at lines 405-407), however, it first refers to the United States (line 411) and subsequently speaks of a model developed in four phases without indicating any government unit (Dubai ???).
- Blockchain and Digital Governments
Line 475: “Some of the proposed projects for blockchain ". These projects are described very briefly just in Table 2, but due to the importance that a model can play, on a large scale, in the analysis and evaluation of empirical evidence, the authors are asked to describe the important characteristics, in the context.
Lines 485-487: The statement reported by the authors is very important, it does not contain bibliographic references. Please enter them.
- Empirical Analysis
Paragraph 7 is very interesting, however the solution and the use case, from a technical point of view, are only very briefly described. In the sub-paragraph "7.1.4 Technical Architecture" the authors report the choices of the protocols adopted (Ethereum, Hyperledger , etc.) or of the interfaces and programming languages, providing only a simple list and not a description of possible architectural solutions to be to be able to adopt in e-government processes and with what specific objective (reliability, incorruptibility, immutability, security, etc.). In the sub-paragraph “7.1.5 Solution Components” the architectural solution is described in a too synthetic form in table 3 and hardly understandable even for a very expert reader of the subject. Authors are advised to structure these two sub-paragraphs differently.
The lack of such technical specifications does not provide the reader with the indispensable elements to be able to evaluate the reliability of the empirical results and the benefits listed in paragraph 8.
- Results of Empirical Study
Paragraph 8 is also very interesting, however the most interesting part is reported in table 4 which is hardly illustrated to the reader. Authors are advised to structure this part of the paragraph differently.
- Conclusions and Future Research
Eliminate questions Q1, Q2, Q3, Q4 (even repeated) by reformulating the paper, as previously mentioned.
The sub-paragraphs " 9.4 Recommended policy actions " and " 9.5 Future research " do not have any relevant scientific content.
- English language
A thorough and careful revision of the English language is recommended. Particolar attention must be paid to grammar form and vocabulary repetition.
Author Response
Dear Editor,
Please find enclosed our revised manuscript, entitled " Blockchain for Governments: The case of the Dubai Government". We are grateful to the editors and peer reviewers for their thoughtful comments that helped us improve the quality of our manuscript.
We answered all the reviewers’ comments and modified the manuscript accordingly. Please find the attached document addressing the reviewer's comments. (Both Reviewer 1 and Reviewer 2)
We look forward to hearing from you regarding our submission and to responding to any further questions and comments you may have.
Yours sincerely,
Authors.

Round 2
Reviewer 1 Report
The authors have address an important guidelines on adapting blockchain technology , they selected the UAE governement as their test case.
The auhors claimed about interviews from experts but doesnt includ in evidence. the validity of guidelines must be made also through experts
Important point is that study uses an active case study done by experts of UAE government thus their permission as well as due credit is required to share their excellent experience.
Author Response
Dear Reviewer,
Thank you for your feedback.
Please see the attachment for responses and actions taken.
Regards,
Authors.

Reviewer 3 Report
I really appreciated the way the authors edited the paper. The insights included by the authors contain acronyms that are neither detailed nor anticipated as a concept. Please specify the following acronyms: Line 114.NHS Add the full terms of the acronym. Line 135 DLT. Add the full terms of the acronym. Line 165 JCR science. Add full terms of acronym. Line 210 Delete character "-" in text "They are - L". Line 229 . AI. Add full terms of acronym. Line 256 DAO attack . Add full terms of acronym. Line 296 DED . Anticipate what is written for the acronym on line 302. Lines 459-464. Note the different font. If the acronym IT is widespread in the paper (also in Table 2) specify it at the beginning. A re-reading of the entire paper is recommended.Author Response
Dear Reviewer,
Thank you for your feedback.
Please see the attachment for responses and actions taken.
Regards,
Authors.
